# Cellular and Molecular Signaling as Targets for Cancer Vaccine Therapeutics

**DOI:** 10.3390/cells11091590

**Published:** 2022-05-09

**Authors:** Wen-Chi Wei, Lie-Fen Shyur, Ning-Sun Yang

**Affiliations:** 1National Research Institute of Chinese Medicine, Ministry of Health and Welfare, Taipei 112, Taiwan; jackwei@nricm.edu.tw; 2Agricultural Biotechnology Research Center, Academia Sinica, Taipei 115, Taiwan; jaclyn@gate.sinica.edu.tw; 3Ph.D. Program in Translational Medicine, College of Medicine, Kaohsiung Medical University, Kaohsiung 807, Taiwan

**Keywords:** cancer immunotherapy, DC-based cancer vaccine, immunogenic cell death (ICD), tumor microenvironment, phytochemicals

## Abstract

Plenty of evidence has recently shown that various inflammatory activities at the local tissue, organ, or even the whole body (systemic) level are strongly linked to many life-threatening chronic diseases, most notably various cancers. However, only very limited information is available for making good use of our supporting immune-modulatory therapeutics for the treatment of cancers. This may result from a lack of studies on specific remedies for efficacious control or modulatory suppression of inflammation-related cancerous diseases. Our group and laboratories were fortunate to have initiated and consistently pursued an integrated team-work program project, aimed at investigating selected medicinal herbs and the derived, purified phytochemical compounds. We focused on the study of key and specific immune-signaling mechanisms at the cellular and molecular levels. We were fortunate to obtain a series of fruitful research results. We believe that our key findings reported herein may be helpful for proposing future thematic and integrated research projects that aim to develop future phytochemical drugs against cancers. The mechanisms of the cellular and molecular systems involved in inflammation are becoming increasingly recognized as keystones for the development of future therapeutic approaches for many chronic and cancerous diseases. Recently, the immune checkpoint inhibitors such as antibodies against PD-1 and/or PD-L1 have been shown to be too expensive for general clinical use, and their effects far from optimal, often showing little or no effect or only short-term efficacy. These results point to the need for developing future immune-regulatory or modulatory therapeutics.

## 1. Introduction

Around thirty years ago, for around a decade, cytokine therapy and cytokine gene therapy using IL-2, GM-CSF, and TNF-α were actively investigated for certain cancers, but the various clinical results were eventually shown not to be useful or applicable [1,2,3]. The difficulty was their action on cytotoxicity and other side effects. Therefore, these days, this approach appears not to be emphasized much. On the other hand, recent findings have shown that progression and metastasis of various cancers are strongly affected by the tumor microenvironment, specifically the (stromal) tissues surrounding the malignant cancerous cells [4,5]. Therefore, appropriate control of innate immunity against cancer progression, the regulation (not limited to inhibition) of inflammation, and the associated/related immune responses are being recognized to have the potential to play an important role in tumor growth and metastasis.

Around 20 years ago, we had the opportunity to establish a mission-oriented, biotechnology research institute in Academia Sinica in Taiwan, and we seized the chance to initiate a phytomedicine program project there. Through our practice as a team on experimental research design and execution, we soon developed a disciplinary culture of pursuing careful planning and coordination as a “team research protocol”. This team effort was established with the following actions in general sequence: (1) Carefully define and set up specific experimental goals, e.g., pick only very limited immune cell inflammation markers (1 to 2 molecular or cellular types) as the only focus for the study aim. Similarly, employ or home-generate only the mouse models that have high relevance to human clinical cancers. (2) Review, rationalize, and clearly identify the target immune cells or molecules that are contemplated in recent reviews to play a key role in cancer inflammation activities. Such candidates were selected from the very recent reports of high impact publications. These examples include immature dendritic cells (iDC), granulocytic myeloid derived suppressor cells (gMDSC), Th17 cells, and the tumor stromal fibroblast cells. For cytokines and chemokines, IL-25, IL- 12, IL-23, IL-17, and others were specifically tested in our study. (3) Employ the newly available technological and experimental systems, e.g., functional genomics, proteomics, tumor stromal microenvironment co-culture systems, and immune cell-based tumor vaccine assays. (4) Explore analytically the crosstalk and networking pathways for immune cell and molecular signaling systems. And (5) Generously share the information and reagents of various home-modified cell lines, the phytochemicals, newly identified chemical structures, and various test reagent preparations. 

Retrospectively, we collected and assembled five key research laboratories, and arranged ourselves in a thematic research fashion. We focused on mechanistic studies of immunity-related phytomedicines, including single phytochemical compounds and the medicinal plant extracts they are derived from. We focused mainly on breast cancer and chronic inflammatory diseases (diabetes and colitis). Over a period of more than 20 years of collective effort, we have obtained good and useful findings. Our results from these studies, as we found recently, could be highly pertinent for integration into an experimental approach system with knowledge-based theory support [6]. We believe strongly that the organized and coordinated team effort mentioned above contributes effectively to our fruitful research results. The various experimental findings may be assembled into a research theme or system, or a school of thought. In this article, we intend to organize and present our recent findings thematically. We thus attempt to create a working hypothesis for future research and development into novel drugs or therapeutics. The physiological target will be the control and regulation of inflammation, and the experimental research will be the pursuit at the cellular and molecular level, aiming for the development of novel phytomedicines/drugs as therapeutics against cancers and specific chronic inflammatory diseases. 

We define the word phytomedicine here as the completely purified single chemical compound (some as commercially available), or highly purified chemical compound(s), as a fractionated chemical group. This phytochemical fraction may contain a more than 94% pure single compound, or it may consist of two to three chemically-related compounds, all tested to ensure they contain no contaminating cytotoxic compounds. Some of these fractions were obtained through a process of bio-organic solvent fractionation followed by HPLC or other purification procedures. The above definition allows the possibility to seek approval for possible new drugs from our study under the “guidelines for botanical drugs of the USA”. 

## 2. Why Target the Inflammatory Cells/Molecules of Immune System for Study of Cancerous Diseases? 

During the past two decades, a series of multi-approach studies have demonstrated that tumor growth, progression, and metastasis often heavily involve the inflammation process [7,8]. Many cancers were shown to arise from the site of tissue damage (e.g., pathogen infection, physical or chemical insults), chronic irritation, and inflammation [7,9,10]. There is also plenty of evidence indicating that various inflammatory cells are routinely but variably present in the tumor stromal microenvironment, and they can play a key role in regulation and modulation of the invasion and metastasis activities of cancer cells. In such tumor microenvironment conditions, various signaling molecules (e.g., cytokines, chemokines) and immune cells (macrophages, dendritic cells, T cells) were shown to participate in critical regulatory or control activities/functions. Therefore, cancer cells and the surrounding inflammatory cells, as well as molecular signaling systems, are increasingly recognized to have a very strong link for coordination in tumor progression and metastasis.

Accordingly, over time our research as a team effort has become more and more focused on immune, inflammatory, and cancerous diseases. In many Asian cultures (especially in Taiwan and China), the community as a whole, along with traditional medicine practitioners, strongly believe that various diseases, tissue-wounding, and pathological fatigue are caused by or associated with a concept expressed in vernacular Chinese that can be translated as “inflammation”. Therefore, according to received wisdom, in order to recover from or heal from such malaise, we need to address and restore the immunity and homeostasis at the organ or holistic level. Therefore the strategy for attacking a spectrum of illness and wound-healing is to find ways and means to treat the inflamed tissue, organs or whole body systems. This notion, rooted in Traditional Chinese Medicine (TCM) is, in fact, quite consistent with the current knowledge embodied in Western scientific medical research. Under this premise, understanding the possible alignment between the Eastern traditional medicine and the current Western medicine, we initiated and engaged in thematic research focusing on anti-inflammatory or immune-regulatory mechanisms for chronic, immune disorders, and cancerous diseases.

## 3. Why Employ Cancer Vaccine Systems Using Dendritic Cells/Tumor Cell Lysate?

The recent development of medical biotechnology to combat cancer metastasis has resulted into two main strategies, namely targeting the immune check point inhibitors (including antibodies against PD-1, PDL-1 and others), and alternatively, the activation of tumor cell lysate-mediated, immune cell-based systems. The latter, as a cancer vaccine system, often involves the effective employment of dendritic cells. Our laboratory in Academia Sinica, Taipei, performed a series of experimental studies [11,12,13,14], aiming to develop phytochemical-treated (PT) tumor cell lysate (TCL) and PT/TCL-pulsed (briefly co-cultured) and DC-based cancer vaccines for use against metastatic tumors. As seen in Figure 1, the experimental system makes good use of the iDC cultures. The specific iDC/tumor cell lysate preparation can be custom-made for the patient’s own clinical needs, and the phyto-extracts or the derived phytochemicals are selected for our study or for anti-cancer clinical interest. For in vivo studies, we have used mainly the metastatic, non-immunogenic 4T1 mouse mammary tumor system, although B16 melanoma and the LNCaP and PC-3 prostate carcinoma systems were also used, resulting in a desirable “comparative experimental test system”.

By going through a serious of sequential, step by step, logically designed animal model studies, our team at Academia Sinica was able to demonstrate a series of important and interesting observations. 

## 4. Some Examples of Key and Potentially Correlatable Experimental Findings Are Shown as Follows

### 4.1. Shikonin Exhibits Various Key Molecular and Cellular Activities and Can Confer Multi-Faceted Medicinal Effects 

The phytochemical shikonin, as a nathoquinone, was shown in our studies to confer a series of highly specific biochemical activities. These include the blocking of transgenic and indigenous TNF-α and GM-CSF promoter activities [15], and later, the suppression of the splicing activity of pre-mRNA via interference with the specific 3′UTR sequence of specific cytokine genes [16]. Shikonin was then shown to regulate the epithelial-mesenchymal transition (EMT) effect [17]. It is also able to upgrade the RANTES activity, resulting in the activation of a gene-based cancer vaccine activity [18]. Functional genomic analysis revealed that 10 or more functionally recognizable genes were significantly up-regulated, and a similar set of different genes were down-regulated [19]. These results indicate that shikonin is able to effectively affect a number of physiologically important immune activities [17,18,19]. 

We then demonstrated that most of these molecular activities result from the binding of shikonin, as a small molecular weight phytochemical, to a very specific nucleotide sequence with structural features of DNA, RNA, and micro RNA molecules. These protein-polynucleotide sequence binding activities were shown to be highly specific for the heterogeneous nuclear RNA-binding protein A1 (hnRNP A1), and this RNA processing activity of shikonin could therefore contribute effectively to the multi-faceted, molecular, and cellular activities in various mammalian systems, in vivo and in vitro [20], as seen by the mechanism shown in Figure 2. 

For instance, shikonin was shown to be highly effective as a key component of a cancer vaccine formulation for the treatment of tumor cell lysate (TCL)-pulsed, dendritic cell (DC)-based tumor vaccines [11]. Many of these activities were later shown to involve the DAMP signaling pathway [11,12,13] and/or the necroptotic autophagy activities [13].

### 4.2. Cytopiloyne and the Associated Polyacetylene Glycosides (PAG) Show Cellular Immune Activities

Cytopiloyne and its two polyacetylene glycoside derivatives were originally extracted and isolated from a medicinal plant, *Bidens pilosa* [21]. The single compound alone or as a mixture was originally shown to confer potent anti-metastatic activity against mammary carcinoma in mice [13,22]. Wei et al. [23] then showed that some of these cellular immune activities were conferred by the regulation of IL-12 and IL-23 activity via a specific interference of the mTOR complex 2 activity.

Initially, a key study on the effect of cytopiloyne on development of diabetes was reported by Yang et al. [21,24]. His group showed that cytopiloyne was able to inhibit CD4^+^ T cell proliferation, and thus effectively prevent the progression of diabetes in non-obese diabetic mice. A normal level of blood glucose and insulin was detected in test animals, and this was also accompanied by normal pancreatic islet architecture. Cytopiloyne not only suppressed the differentiation of type 1 Th cells but it also promoted that of type 2 Th cells. These results are consistent with the finding on enhancing GATA-3 transcription. Furthermore, long-term treatment with cytopiloyne effectively reduced the level of CD4^+^ T cells presented in the pancreatic lymph nodes and spleens. However, this cytopiloyne treatment did not compromise the total Ab responses mediated by T cells. Co-culture experiment showed that this decrease in CD4^+^ T cells involved the Fas ligand/Fas pathway. These findings together suggest that cytopiloyne can effectively prevent type 1 diabetes, most likely mediated via a T cell regulation mechanism. 

### 4.3. Expression of IL-23 and IL-12 in Dendritic Cells Can Be Effectively Regulated by Mammalian Target of Rapamycin Complex 2 (mTORC2) 

IL-12 p40 is known to play a key role in the development of Th1 and Th17 cells and specific autoimmune diseases. It is also a common subunit for both IL-12 p70 and IL-23. Efficacious regulation of IL-12 p40 expression has therefore been contemplated as a useful strategy for developing therapies against Th1- and Th17-mediated autoimmune diseases. The mTOR protein has two subunits, mTORC1 and mTORC2. The mTORC1 subunit has been shown to mediate IL-12 p40 expression in dendritic cells and the relevant signaling molecules. However, little to no information was available about the role of mTORC2 in IL-12 p40 expression. 

Wei et al. [23] demonstrated that cytopiloyne can effectively inhibit mTORC2 activity, resulting in specific inhibition of LPS-induced expression of IL-12 p70, IL-23, and IL-12 p40 in test human DCs (Figure 3). Regulation of this mTORC2 activity by cytopiloyne was shown to involve Akt activation and a persistent phase of NF-κB activation. In terms of IL-12 p40 expression, these findings revealed a new role for the mTORC2 pathway which is antagonistic to the activity of mTORC1. Wei et al. [23] hence provided us with a new understanding of the mTOR regulation of IL-12 p40-mediated Th1 (IFN-γ) and Th17 (IL-17) responses. Their results suggest that the phytochemical cytopiloyne may be further investigated for future applications as therapeutics for Th1 and Th17 cell-mediated autoimmune or inflammatory diseases.

### 4.4. Phytochemical Polyacetylenes Inhibit Differentiation of MDSCs from Bone Marrow and Impair Tumor Metastasis

Myeloid-derived suppressor cells (MDSCs) have been strongly implicated in the promotion of tumor metastasis through protecting metastatic cancerous cells from immune surveillance. These cells have thus been suggested as novel targets for cancer immunotherapy. We demonstrated previously [22] that subcutaneous treatment with certain polyacetylene glycosides (BP-E-F1), as a plant extract fraction derived from the medicinal plant *Bidens pilosa*, can effectively suppress 4T1 mouse mammary tumor metastasis. At the cellular mechanistic level, BP-E-F1 can inhibit the tumor-induced accumulation of gMDSCs. Furthermore, the study demonstrated that these compounds exert anti-metastasis activity by inhibiting the differentiation and function of gMDSCs in test mice. Mechanistic and pharmacokinetic studies revealed that BP-E-F1 suppressed the differentiation of gMDSCs via effective inhibition of the stromal tumor-derived, G-CSF-induced signaling pathway in the bone marrow cells of test mice. Taken together, these findings showed that a group of plant polyacetylene glycosides could specifically target gMDSC differentiation from the bone marrow cells [22]. This finding can be considered for potential future application of BP-E-F1 as a candidate botanical drug against metastatic breast cancers. 

### 4.5. The Effect of Deoxyelephantopin and Derivative DETD-35 on Lung and Breast Cancers 

The bio-efficacy of a plant sesquiterpene lactone, deoxyelephantopin (DET) and its derivative, DETD-35, was studied in metastatic melanoma cells in vitro and in vivo in a xenograft mouse model [25]. Treatment with both phytochemicals inhibited A375LM5^IF4g/Luc^ tumor cell proliferation, as evidenced by molecular markers for G2/M cell-cycle arrest and apoptosis activities. DET and DETD-35 induced reactive oxygen species (ROS) generation and glutathione (GSH) depletion in test cells. Mitochondrial damages, superoxide production, and mitochondrial bioenergetics dysfunction also occurred. In accordance, DET and DETD-35 effectively inhibited lung metastasis of A375LM5^IF4g/Luc^ tumors in NOD/SCID mice. Mechanistically, this result was shown to occur through the inhibition of tumor cell proliferation, angiogenesis (VEGF+, CD31+), and EMT (*N*-cadherin) activities of the tumor microenvironment in mice lungs. These findings indicate that DET and DETD-35 may potentially be employed as therapeutics for future intervention in lung metastatic carcinomas [26].

DET and its novel derivative were also shown by Nakagawa-Goto et al. [27] to suppress potently human triple negative breast cancer (TNBC) MDA-MB-231 tumor growth in test mice. The mechanism underlying the activity of DET and DETD-35 was induction of ROS which caused paraptosis-like cell death [28] and structural and functional damage to mitochondria [29]. This subsequently evoked exosome release from test mammary carcinoma cells. Intriguingly, this exosome activity, induced by both compounds, had an atypical function. Cancer cell-derived exosomes were previously shown to confer increased metastatic potential. But in this case, after DET/DETD-35 treatment, the released exosomes showed anti-proliferative activity against MDA-MB-231 cells. Proteomic analysis of test TNBC cells showed that DET and DETD-35 effectively attenuated the expression of a group of specific proteins, including those related to cell migration/cell adhesion and angiogenesis. In addition, a number of proteins that participate in cellular and molecular mechanisms, including oxidative stress and transmembrane potential in mitochondria, were found deregulated by treatment with these compounds. This study [29] therefore showed that the phytochemical DET and its analog DETD-35 can effectively inhibit TNBC cellular activities, apparently via mediating oxidative stress-induced exosome release, in tandem with an alteration of the composition and function of specific exosomal proteins. The findings of this study thus suggest that DETD-35, as a novel DET derivative, has potential for further development into anti-TNBC therapeutics. This research on DET and DETD-35 again suggests the multi-faceted effects of these phytochemicals on various cancerous diseases. In addition, this observation adds support for thematic research and the development of anti-inflammatory phytochemicals and the medicinal plants that they are derived from for use against cancer and immune-related chronic diseases/disorders.

### 4.6. Q2-3, a Plant Lignin-Derivative, Can Induce Endogenous IL-25 in Tumor-Stromal Fibroblasts, Resulting in Strong Anti-Metastasis Effect 

Q2-3 is a methyl ester dimer of caffeic acid that is a very common plant secondary metabolite. Q2-3 is therefore considered as a plant lignin derivative. We studied the mammary tumor microenvironment and demonstrated that Q2-3 may have translational medicine potential. When tested in vitro using a co-culture system with fibroblast-conditioned medium and target tumor cells, we showed that Q2-3 can very effectively induce the functional activity of IL-25 secretion from the tumor stromal-associated fibroblast cells, and subsequently activate specific T cells or other immune cell types, resulting in a strong anti-metastasis activity in vivo in a test mouse model (Figure 4). These findings suggest the potential efficacy of an approach employing strong, non-cytotoxic, tumor micro-environment (stromal cell)-regulatory cytokines, and specific immune cell activity, for a highly potent attack against tumor infiltration or metastasis [30]. 

### 4.7. LAW Has Multiple Effects in Suppressing Prostate Cancer Growth and DSS-Induced Colitis Activity in Mouse Models 

LAW is the acronym used to describe luteolin, apigenin, and wedelactone in a molar ratio combination as presented in a specific medicinal herb, *Wedelia chinesis*. This medicinal plant is traditionally commonly used as a herbal medicine and is still very much in use in Taiwan today. Previous studies by Lin et al. [31] showed that the phytocompound LAW, as a HPLC fraction in a molar ratio as that similarly present in their medicinal plant of origin (*W*. *chenesis*), could synergistically suppress androgen receptor activity in the tested prostate cancer cells, in a biochemical analysis system. It was then shown, via oral intake, to attenuate the orthotopic growth of prostate cancer in a nude mouse system [32]. The group further demonstrated that this anti-tumor activity was mediated via an IKK signaling pathway [31]. These studies thus provided us with highly specific and integrated results on the anti-prostate cancer activity of LAW. The phytochemical LAW induces suppressive activity on Th1 cytokines and IL-17 expression and signaling, and confers strong anti-colitis activity in mice [33]. We recently showed that LAW can confer a strong anti-inflammatory effect in mice. In a DSS-induced mouse colitis model, we showed that the Th1 cytokines including TNF-α and IFN-γ were induced to high levels [33]. Similarly, as was also shown with the IL-17 cytokine, upon oral feeding of test mice with the LAW phytochemicals, these inflammatory activities were shown to be greatly reduced, in parallel with a strong reduction of the severity of colitis activity in test mice [33]. This effect was further studied and was attributed to a master blockade of the IL-17A signaling in vivo [34]. 

### 4.8. An Echinacea Purpura Phyto-Extract Fraction (BF/S+L/Ep) Regulates Several Cytokine and Chemokine Genes in Human Dendritic Cells

In this study, the specific and differential gene expression of key immune cell and signaling systems [35,36] were investigated in human immature dendritic cells (iDC), using Affymetrix DNA micro-array analysis. It was revealed that several genes for cytokines (IL-8, IL-18 and IL-1β) and chemokines (CCL2, CCL-5 and CXCL5) were up-regulated in test human iDCs within 4 h of BF/S+L/Ep treatment. Bio-informatics analysis further revealed a key signaling network involving the various immune-modulating molecules led to the activation of a downstream molecule, the adenylate cyclease 8 (AC8). AC8 and cAMP are known to modulate a number of important metabolic and biochemical activities of most mammalian cells. Subsequently, our proteomics study [35] demonstrated that, at 12 h post BF/S+L/EP and cichoric acid (as key component of this plant) treatment, there was a strong up-regulation of an anti-oxidant and anti-inflammatory protein activity, and the superoxide dismutase (SOD) was detected. These findings regarding *Echinacea purpura* give useful support to the notion that this medicinal plant, long-believed to be a medicinal plant in many Western cultures, may indeed have some immuno-modulatory activities at the cellular and tissue levels. 

## 5. Specific Phytochemical-Treated Tumor Cell and Immune Cell Vaccines for Future Clinical Therapeutics 

Traditional herbal medicines have been historically used for more than three thousand years, most notably in China, India, and, more recently, in Germany. In 1995, the Nobel Prize in medicine and physiology was awarded to Professor Too You You of China, for her contribution in fighting malaria, by the identification and use of phytochemical artemisinin from the Artemisia plant, which has been commonly used in traditional Chinese medicine (TCM). This award recognition highlighted the past and the current importance of TCM and other traditional herbal medicines. Nonetheless, the possible biochemical mechanisms and the resulting pharmacological activities were only demonstrated and published years later in a J. Exp. Med article [37]. As a result, it has become increasingly appreciated that medicinal plant extracts and the derived phytochemicals, phytocompounds, could have highly specific biochemical activities that can confer bona fide pharmacological effects. On the other hand, it is well known, from TCM practitioners, that a series of human inflammatory diseases, including those for skin, stomach, liver, and other organs, could be and have been actively treated by using similar types in formulation as herbal medicine prescriptions. Such a traditional use of a group of relatively limited herbal species as plant source materials points to a common feature on the capacity for treating or therapy against various anti-inflammatory diseases. We consider our current studies have provided us with useful information and good rationale for future systematic studies on specific phytomedicines, singularly or in combination with other therapeutic agents.

We have previously rationalized a potentially useful laboratory vaccine approach. Now, after a series of experimental studies providing promising results and findings through the past 15 years, we now believe strongly that we may indeed have created a new clinical cancer therapy strategy, as shown in Figure 1. This schema of immunotherapy for cancer makes good combinational use of the PC/TCL/DC vaccine preparation, consisting of specific phytochemicals, the patient’s own tumor cell lysate, and the patient’s autologous dendritic cell or other stromal immune cell preparations, such as a cell-based cancer vaccine system. We hope that this cancer vaccine treatment could be effectively employed, sooner and not later, for future therapy of metastatic cancers. 

## 6. Conclusions

As described above, using a series of experimental studies conducted in our institute in Taiwan, we have been able to fruitfully obtain systematic and thematic research results on anti-inflammation and immune-modulation related diseases, including specific cancers. Whereas we appear to have been fortunate in defining and approaching these studies in a coordinated “program project” format, we also consider that we have established a research strategy with a specialty on clearly setting up specific aims and a consistent pursuit of step-by-step, sequential laboratory progress for a medicinal plant research program, aiming at developing novel botanical drugs. This R&D strategy is represented as a flow chart shown in Figure 5. This comparative analysis shows the important difference of pursuing two very different experimental pathways, i.e., the “conventional” versus the Agricultural Biotechnology Research Center (ABRC) approach. In our current ABRC approach, we need to firstly perform specific and multiple anti-inflammation assay or tests, instead of the conventional method of performing cytotoxicity tests, which is routinely practiced by virtually all pharmaceutical R&D companies or institutes. Secondly, when candidate phytochemicals are obtained, we need to immediately try to perform in vivo experiments of these candidate lead compounds for the target disease of our interest (e.g., breast, prostate or melanoma cancers, colitis, or diabetes) by using well-recognized in vivo animal tumor or disease models. With good positive results that we obtained in this sequence, we can then investigate the pharmacological effect by using various experimental “omics” approaches. Finally, we can then evaluate the immune-signaling and -mechanism activities at the cellular and molecular levels. Our productive research results led us suggest that our approach shown in Figure 1 could in fact be integrated and assembled within or among different institutions as collective efforts, aiming for developing systematically novel, phytochemical-derived anti-cancer drugs.

## Figures and Tables

**Figure 1 cells-11-01590-f001:**
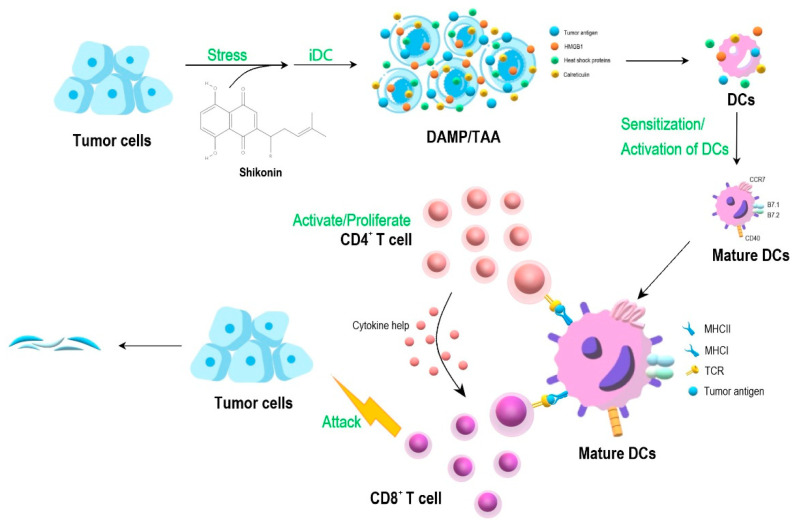
A DC-based cancer vaccine approach. Effect of shikonin-induced ICD in tumor cells for activation of DCs, resulting in T cell activation and enhanced attack tumor cells. Abbreviations: DAMP, danger associated molecular patterns; TAA, tumor associated antigens; TCR, tumor cell receptor; CD4^+^ T, CD4^+^ type thymocyte cells.

**Figure 2 cells-11-01590-f002:**
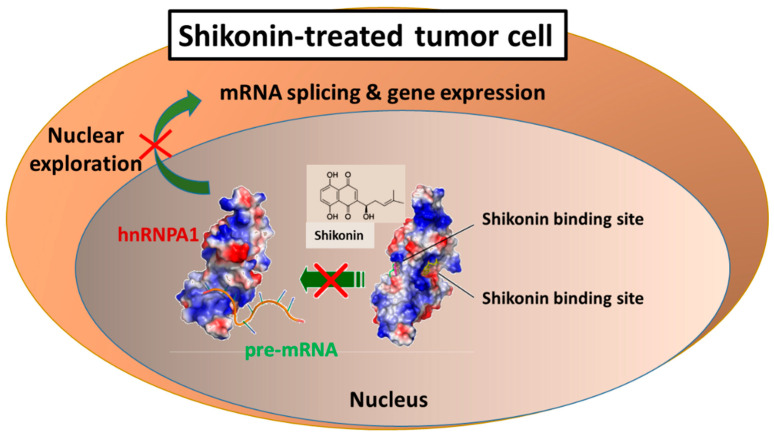
Identification of hnRNPA1 as a protein target provides pharmacological basis for diverse biochemical activities of shikonin. Docking models of the Shikonin binding to hnRNPA1, generated by AutoDock Vina and presented by a molecular visualization program, PyMOL.

**Figure 3 cells-11-01590-f003:**
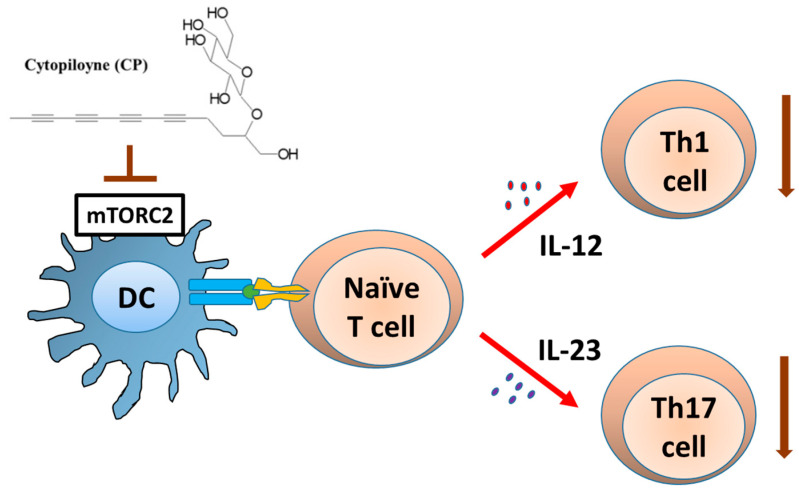
Immunomodulatory activity of cytopiloyne (CP). Blocking mTORC2 activity by CP can specifically inhibit LPS-induced expression of IL-12 and IL-23 in human DCs, resulting in the suppression of Th1 and Th17 activation.

**Figure 4 cells-11-01590-f004:**
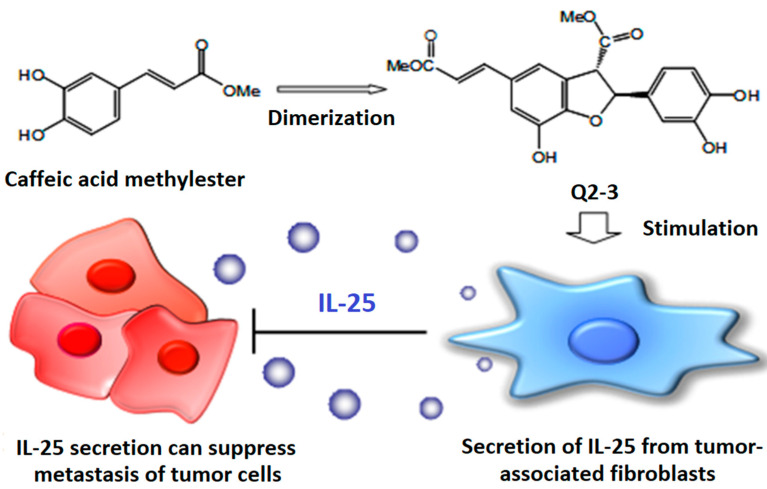
Anti-metastatic mechanism of methyl(E)-3-[2-(3,4-dihydroxyphenyl)-7-hydroxy-3-methoxycarbonyl-2,3-dihydro-1-benzofuran-5yl]-prop-2-enoate (Q2-3). Q2-3, the dimerization product of plant caffeic acid methyl ester, suppresses 4T1 metastasis by increasing fibroblastic IL-25 activity.

**Figure 5 cells-11-01590-f005:**
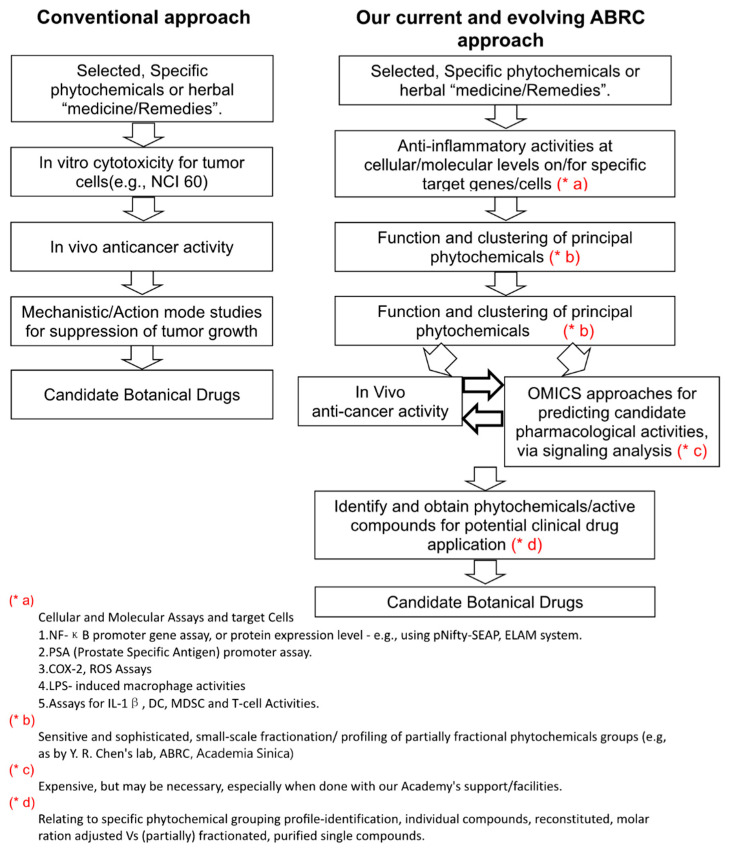
Emerging approaches for ABRC phytomedicine research strategy for botanical drug development.

## Data Availability

Not applicable.

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
