# Peer review of "Cellular and Molecular Signaling as Targets for Cancer Vaccine Therapeutics"

_cells, 2022, doi:10.3390/cells11091590_

Round 1
Reviewer 1 Report
In this article, the authors comment on some molecular and cellular signaling toward targeting cancer therapy, with focus on the use of dendritic and tumor cells, based on their group/laboratories findings.
Comments:
1- I think the title needs to be modified. I guess it would be much better to change “Cancer Immunotherapy” by “Cancer Therapy Using Dendritic/Tumor Cell Lysate” or “Anti-Inflammatory and Immune-Modulation Related Cancer Diseases”.
2- Dendritic cells (DC), in line 126, was already introduced in line 56. Similarly, “the DC immune cell culture” (line 130) should be “iDC cultures”
3- It would be better to delete “as follows”, which exists at the end of page 3, as directly after that there is the section 4., which the subtitle finishes already with “as follows”.
4- In figure 1, what all the reported abbreviations stand for? Full names should be provided as footnotes, specifically for the non-common ones. The readers will be lost, particularly between iDC and ICD.
5- Reference no. 19 should be translocated to the end of the sentence; directly after “…down-regulated” and “My laboratory…” should be “ Our laboratory…”. Moreover, the authors should indicate that it refers to the 2nd affiliation, which is cited in the article.
6- Full name of iDC, which figure in line 316 was given in line 56.
7- Section 5 is very short once compared with the other sections. Taken into account that phytochemicals are showing promising therapeutic effects on cancerous diseases including those associating inflammation and/or immune modulation, I think this section should be developed.
8- No need to refer to the Fig.1 twice in the section 5.
9- In line 352, and similarly to my comment 5, it would be better to indicate that’s the 2nd affiliation is concerned. Authors can give the full name of the institution “Agricultural Biotechnology Research Center (ABRC)” instead of “our institutional name”.
10- In figure 5 footnotes, the authors should provide at least the institution to which Chen’s Lab belongs.
Minor comment:
11- Punctuation and spaces should be checked. For example: “IL- 12” should be “IL-12”; “discuss” should be “ discussing”, “29º” should be “29th”,…
Reviewer 2 Report
I have gone through the manuscript. Topic is indeed interesting but it needs to be revised moderately.
Authors should explain in detail about DAMPs.
Authors should explain about metastasis. I will encourage authors to comprehensively analyze underlying mechanisms of metastasis.
How mTOR pathway can be pharmacologically exploited.
Authors have to provide more convincing evidence of PDL1-driven signaling in regulation of carcinogenesis and metastasis.
Round 2
Reviewer 1 Report
Overall, the manuscript has been improved following the recommendations addressed by the reviewers. I think it's acceptable in the current form. Just minor corrections should be done.
- In figure 1 legend, CD4+ is cited twice.
- line 336, the word "been" is missing.
- Some space errors: such as : Med article[37].
Reviewer 2 Report
Looks in acceptable form now.